# DSCONV: DYNAMIC CONVOLUTION ON SERIALIZED POINT CLOUD

## ABSTRACT

In recent years, research on point-based architectures has advanced rapidly, showcasing their competitive performance. However, the unstructured nature of point clouds limits the application of effective operators such as convolutions in feature extraction. Although many works have attempted to address the issues of unstructured data and introduce convolutions or transformers, the complex spatial mappings of point clouds and cumbersome convolution implementations in these methods limit real-time performance of the model. Furthermore, excessive structural mapping ignores the independence of point cloud position representation and fails to capture finer-grained features. To tackle these challenges, we serialize point clouds to provide them with structure and introduce AdaConv to directly utilize 2D convolutions, which simplifies the process and better preserves the relative positional relationship. Additionally, we propose a novel dynamic refinement approach for point cloud positions, continuously modifying the coordinates of points within the convolutional neighborhood to enhance the flexibility and adaptability. We also integrate local and global features to compensate for the loss of point cloud features during downsampling. Finally, we propose DSConv based on PointNeXt, maintaining scalability and inference speed. By combining DSConv with new architectural designs, we outperform the current state-of-the-art methods on ScanObjectNN, Scannet V2, and S3DIS datasets.

## 1 INTRODUCTION

In the realm of dense point cloud analysis, there are currently two main approaches: one represented by models like Qian et al. (2022) and Thomas et al. (2019), which utilize MLP and convolutional structures; the other employing self-attention mechanisms (Guo et al., 2021; Zhao et al., 2021; Wu et al., 2022; 2024) for point cloud analysis. The former relies on unstructured data or complex mappings, where the MLPs focuses on point-wise feature analysis, while point cloud convolutional structures perform complex equivalent convolutions over neighborhoods, failing to achieve the fine-grained neighborhood feature extraction based on spatial positions like 2D convolutions. The latter struggles to effectively capture local features. Hence, neither method fully leverages the local relative positional relationships among points. Redundant MLP computations and complex 3D convolutions also introduce computational burdens. Moreover, the richer degrees of freedom and structure in three-dimensional objects makes addressing issues such as object rotation and deformation more challenging. To address these challenges, we propose a flexible sequential convolution method to extract richer local features with strong real-time performance and leverage the independence of point cloud positions to cope with changes in object posture.

Recent advancements, such as PointGPT (Chen et al., 2024) have addressed the unstructured nature of point clouds by employing serialization, allowing for the extraction of features based on relationships in one-dimensional space. Inspired by this, the core motivation of our work is to utilize serialization techniques to transform unstructured point cloud data into sequences with structured spatial features, followed by the extraction of local features through designed convolution. To explore these methods, we propose **Dynamic Sequence Convolution (DSConv)** built upon Inverted Residual MLP (InvResMLP) module of PointNeXt (Qian et al., 2022) for Local Aggregation. We map points within groups across multiple directions and compute sliding windows along the corresponding sequence to extract local features, which is a fine-grained convolution without complex mappings, while improving the reduction method to ensure global features. We couple point posi-

tions with convolution parameters to enable dynamic adjustments of these parameters based on the distribution of points within the convolution neighborhood. Additionally, we propose an efficient method for point position refinement to handle target pose variations, dynamically adjusting point positions based on the features of neighbors to enhance flexibility while maintaining high throughput. Specifically, our contributions are summarized as follows:

- We introduce point cloud serialization and adaptive convolution (AdaConv), which replace the reliance on unstructured point clouds or complex spatial mapping, enabling better utilization of relative positional relationships among point clouds and less computational burden.

- We propose a dynamic point position refinement method to address challenges with object pose variation in three-dimensional space.

- We adopt a local-global structure that enables network to flexibly combine global MLP and local features, thereby reducing instability and feature loss in traditional models. We also couple point cloud positions with convolution parameters to improve the generalization of DSConv.

- Our proposed DSConv shares the same scalability and inference speed as Qian et al. (2022), with fewer parameters. DSConv-XL outperforms PointNeXt-XL by a 4.8% mean IoU (mIoU) increase on S3DIS (Armeni et al., 2016) (6-fold cross-validation). Additionally, through further architecture designs, DSConv-XXL achieves state-of-the-art performances across multiple tasks.

## 2 RELATED WORK

**Point-based Networks.** Point-based networks provide greater flexibility and preserve the original information more effectively compared to voxel-based methods (Zhou & Tuzel, 2018; Yan et al., 2018). PointNet (Qi et al., 2017a) introduce MLPs for handling unordered point cloud data. PointNet++ (Qi et al., 2017b) introduce a hierarchical structure for feature aggregation at different scales. Wang et al. (2019b;a); Qian et al. (2021a) utilize graph neural networks for point cloud analysis. Yu et al. (2022); Pang et al. (2022) utilize self-supervised algorithms for pre-training. Transformer-like networks (Zhao et al., 2021; Wu et al., 2022; 2024) extract local features via self-attention. Ma et al. (2022); Qian et al. (2022); Deng et al. (2023) improve model performance through residual connections and different feature preprocessing techniques. However, the use of unstructured point clouds in these methods impedes the exploitation of relative position information, while the serialization and convolution strategies in DSConv effectively utilize this information.

**Convolution in Point Clouds.** Convolution, as one of the effective methods for processing local features, has always been a focus in point cloud analysis. SpiderCNN (Xu et al., 2018) defines convolution kernels as Taylor-expansion-based polynomials, encoding geometric information via step functions within local neighborhoods. PointCNN (Li et al., 2018) learns an X-transformation from input points to reassign associated features and feature weights, enabling convolution to process regularized input features. PointConv (Wu et al., 2019) generates convolution kernels by applying MLPs and kernel density estimation to points, convolving them with neighborhood points. KPConv (Thomas et al., 2019) introduces rigid kernel point deformable convolution, where the parameters of kernel points are used to process the points mapped to that position. These convolution methods in point clouds rely on complex mappings that affect the algorithm speed, while our serialized mapping approach is simpler and reduces the computational cost.

**Utilization of Point Cloud Serialization.** Recent works have begun exploring serialization methods to process raw point clouds, aiming to overcome their inherent disorder and non-structural nature. OctFormer (Wang, 2023) applies Morton-like sorting to octree-based point cloud data and performs windowing along the resulting sequence. PointGPT (Chen et al., 2024) utilizes Z-order (Morton, 1966) sorting on points during self-supervised pre-training, leveraging remaining points to predict missing patches. Point Transformer V3 (Wu et al., 2024) alternates between Z-order and Hilbert curves (Hilbert & Hilbert, 1935) to convert point clouds into patches, aiming to mitigate the impact of K-Nearest Neighbors (KNN) search on speed. These methods suggest that serialized raw point cloud data still maintains a good spatial proximity relationship. However, the aforementioned serialization methods can only ensure compactness within the sequence, with relatively poor ability to preserve relative positional information among point clouds. In contrast, our approach to serialization leveraging positional relationships avoids this limitation.

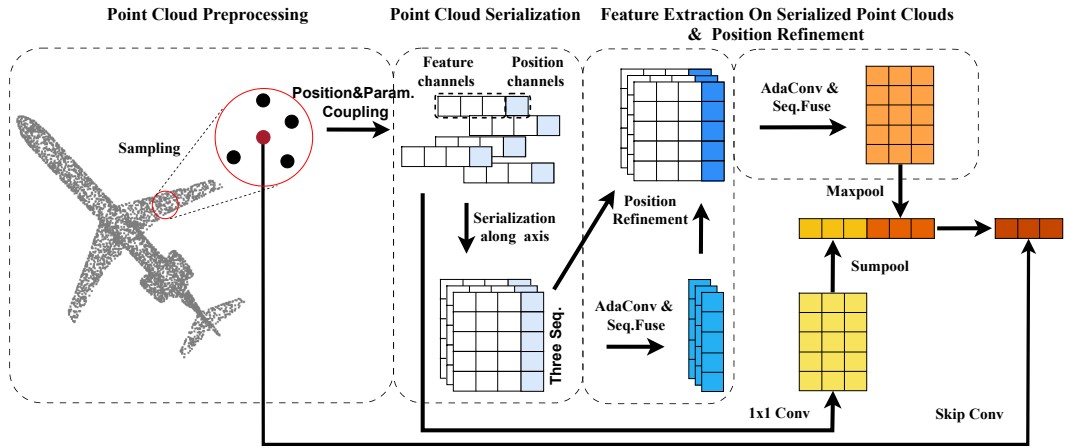

Figure 1: Overview of the DSConv module. The red point serves as the center for feature extraction. All input points are treated as center points once during DSConv module processing.

# 3 METHOD

We propose a local feature extraction method based on serialized point clouds and convolution to enhance the utilization of relative positional information, as shown in Fig. 1. This section covers the serialization of point clouds in Sec. 3.1, the design of feature extraction on serialized point clouds in Sec. 3.2, the introduction of dynamic point cloud position refinement in Sec. 3.3, a description of the coupling between position feature and convolution parameters, along with the integration of global and local features in Sec. 3.4, and the overall structure of DSConv in Sec. 3.5.

## 3.1 POINT CLOUD SERIALIZATION

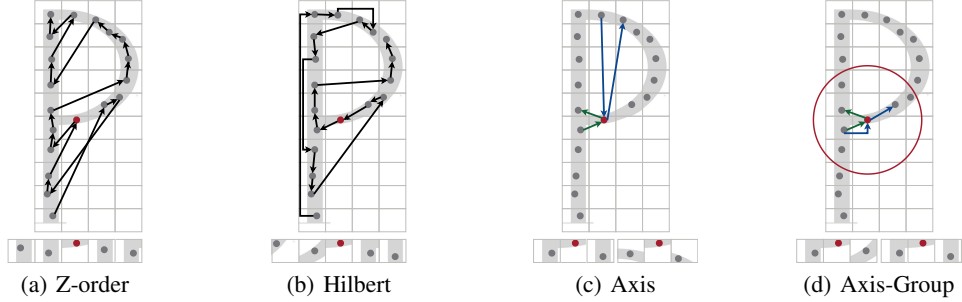

|       (a) Z-order       |       (b) Hilbert       |        (c) Axis        |     (d) Axis-Group      |

Figure 2: The effects of four serialization algorithms on the plane, with red points indicating the center points. A total of 4 points (excluding the center point) are sampled. The lower part shows the feature sequence of the sampled points. (a) Z-order serialization. (b) Hilbert serialization. (c) Serialization along the coordinate axis. (d) Serialization along the coordinate axis with group constraints. The green arrow indicates the sampling sequence along the x-axis, while the blue arrow indicates the sampling sequence along the y-axis. Additionally, the red circle is the range of groups.

Point cloud collection methods and data storage formats often result in sparse and unordered point clouds. To address this issue, point clouds are typically converted into voxels (Zhou & Tuzel, 2018) or pillars (Lang et al., 2019). However, these approaches not only incur significant computational overhead but also lead to the loss of information in dense regions. In contrast, point cloud serialization, while imparting structure, can preserve all point cloud information intact and achieve higher efficiency. By designing appropriate mapping methods, points in multi-dimensional space can be mapped to a one-dimensional continuous sequence, preserving the local relationships between points. We propose a multi-directional point cloud serialization method and compare it with

existing methods. Mapping methods based on distance relationships, such as those shown in Fig. 2a and Fig. 2b, can better ensure the compactness of adjacent points in the sequence, but are limited in representing the relative positional relationships of points. By comparison, our proposed method, based on mapping along three coordinate axes (as shown in Fig. 2c and Fig. 2d), can directly represent the relative positional relationships of points in the point cloud, thereby preserving local relative positional features of objects to a certain extent and providing better geometric representation capabilities. Additionally, our approach does not require encoding, leading to greater efficiency.

In most cases, simply serializing along coordinate axes may map points that are far apart to nearby positions in the sequence (as shown in Fig. 2c). Therefore, we map the point clouds after grouping module, and by leveraging the compactness of the grouped point clouds, we can effectively avoid the issue of distant points being mapped to close positions due to non-encoded mapping. With the constraint of grouping, our serialization method achieves similar results to those based on encoded distance serialization, and the order within each sequence also reflects the relative positional relationships between point clouds, as shown in Fig. 2d. The serialized point cloud, which can directly reflect the spatial physical distribution, simplifies the implementation of 3D spatial convolution and enables finer-grained point cloud convolution.

### 3.2 LOCAL FEATURE EXTRACTION ON SERIALIZED POINT CLOUDS

**Adaptive convolution.** Although serialization structures point cloud data, allowing for the direct application of 2D convolution, there is still a drawback in using convolution for point cloud processing: the point cloud needs to overcome the influence of absolute positions and focus solely on the local positional relationships within the convolution neighborhood. Therefore, we propose an adaptive convolution to address this issue, thereby achieving the translation invariance of convolution and extracting the local features. To ensure that convolution attends only to relative positional information in the neighborhood, we fix certain parameters of the convolutional operator and embed the decentering process into the convolution operation. We modify the center position parameters in the input channels of position coordinates as follows:

$$w_0^{position} = - \sum_{j=-N, j \neq 0}^{N} w_j^{position}, \tag{1}$$

where $w_0^{position}$ represents the parameter for processing the position channel at the convolution center point, $w_j^{position}$ denotes the parameters at other positions of the convolution operator when processing the position channel and $N$ is the size of $1 \times N$ convolutional kernel. Redefining the convolutional parameters according to Eq. 1, the convolution computation, which takes features incorporating positional coordinates as input, is transformed into the following:

$$feature_i = \sum_{j=-N}^{N} (w_j^{position} \times p_{i+j} + w_j^{feature} \times f_{i+j})$$

$$= \sum_{j=-N}^{N} w_j^{feature} \times f_{i+j} + \sum_{j=-N, j \neq 0}^{N} w_j^{position} \times p_{i+j} - \sum_{j=-N, j \neq 0}^{N} w_j^{position} \times p_i \quad (2)$$

$$= \sum_{j=-N}^{N} w_j \times [p_{i+j} - p_i; f_{i+j}],$$

where $p_i$ and $p_{i+j}$ represent the coordinates of the central point and its neighboring point, respectively, $f$ denotes the corresponding feature values, $w^{feature}$ is the convolution parameters on feature value channel, and $feature_i$ is a single convolution result in the neighborhood of the central point $i$. This calculation achieves dynamic decentralization of coordinates during convolution.

**Feature extraction on serialized point clouds.** After grouping and serialization, the point features are represented in a form similar to a four-dimensional tensor of images, allowing for the application of convolutions along the group dimension to perform sliding window calculations and extract features. In order to ensure sufficient receptive fields in the spatial dimensions and reduce memory usage, we propose a multi-sequence fusion method based on AdaConv, enabling processing of features from different sequences, akin to 2D convolutions, as depicted in Fig. 3.

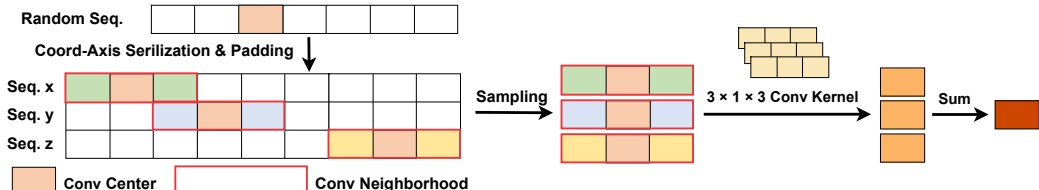

Figure 3: Multi-sequence fusion. After sorting points along three directions, three sets of sequences are obtained. Then, the padded results are convolved using AdaConv, where padding ensures that the lengths of the three sets of sequences remain the same as the original sequence after convolution, which is necessary. For one point in original sequence, different neighborhoods of this point along the three sequences are convolved separately, and then fused into a feature point. This method is equivalent to using a 3×3 convolution to process a 3×3 receptive field.

### 3.3 DYNAMIC POINT POSITION REFINEMENT

The objects in three-dimensional space exhibit richer geometric transformations. To reduce the impact of these intricate pose variations on feature extraction, we propose point position refinement, leveraging the characteristic of position information embedded in the feature channels of point clouds. This method dynamically modifies the position of input points for each group using the convolution introduced in Sec. 3.2, thereby enhancing the stability and consistency of features.

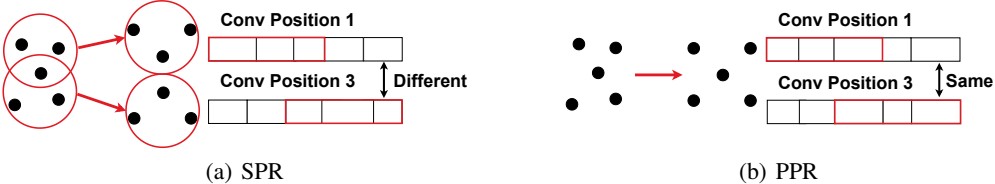

(a) SPR             (b) PPR

Figure 4: Position refinement and convolutions after refinement. (a) Sequential position fine-tuning, adjusting the positions of points within the convolutional neighborhood as needed without modifying the original point cloud distribution. (b) Fine-tuning points within each group individually.

**Sequential Position Refinement (SPR).** We utilize the convolution strategy discussed in the previous section to compute the required offsets for each point within the receptive field when convolving at any position. Then, these offsets are applied to the position channels (as shown in Fig. 4a), enabling different fine-tuning for convolutional inputs at different positions without affecting the original positions. We attempt to implement this operation using Deformable Convolutional Network (DCN) (Dai et al., 2017b; Zhu et al., 2019; Wang et al., 2023). However, DCNs result in significant memory consumption and considerably slow down the algorithm. Therefore, we propose a more efficient method, as shown in Eq. 3&4:

$$\Delta p = Fuse(AdaConv_i(Ser_i([(p - p_{center})/R; f]))), i = x, y, z. \tag{3}$$

$$Output_i = AdaConv_i^p(\Delta p_i) + AdaConv_i^{p+f}([(p - p_{center})/R; f]), i = x, y, z, \tag{4}$$

where $Ser$ denotes the serialization process, $R$ represents the group radius, and $p_{center}$ denotes the center point coordinates. By employing the method mentioned in Sec. 3.2, we generate a offset $\Delta p$ in the number of channels corresponding to a 3×receptive field size, representing the offset of each point in three directions within the convolutional neighborhood. The original input is computed using $AdaConv^{p+f}$, and the offsets are calculated using $AdaConv^p$, which is a part of the parameters of the former, aiming to fine-tune the input positional features. This calculation eliminates the need to generate expanded feature maps, thereby significantly reducing memory usage and improving speed. Additionally, it offers high flexibility and adaptability. Unlike directly modifying the positions of the original point cloud, it can fine-tune the neighborhood points at different positions according to their needs, better adapting to complex spatial structures and morphological changes.

**Point-wise Position Refinement (PPR).** Despite optimizations, the above method still consumes a lot of resources, as it requires storing the coordinate offsets of all points within the convolutional neighborhood at each convolutional position. Therefore, we propose a simpler position refinement method that only calculates the offset of the center point in convolutional neighborhood and then directly modifies its position within the group using that offset, as shown in Fig. 4b.

## 3.4 NETWORK DETAILS

**Local-global structure.** Local Aggregation (LA) structure employs saturation sampling during the downsampling process, resulting in significant redundant computations. Additionally, its max pooling excessively emphasizes the features of individual points within a group, neglecting others and missing out on holistic features. Therefore, we retain the original LA module in DSConv module but replace its max pooling with sum pooling to enhance the speed and improve its ability to extract global features, reducing over-reliance on individual key points. A MLP is used to mix them.

**Coupling position with convolution parameters.** We improve the flexibility of DSConv by coupling the positional information of local point cloud with convolution parameters, providing different computation parameters for points at different locations within the local space. We propose using positional encoding to implement the coupling process, for each channels as follows:

$$w(f + PE(p)) = wf + bias^{coupled}, \tag{5}$$

where $f$ and $p$ represent the features and positional coordinates of each point within each group after coordinate decentralization, respectively. $PE()$ is positional encoding. This approach introduces a bias that couples position with convolution parameters for each channel computation result. Unlike Vision Transformer (Dosovitskiy et al., 2020; Carion et al., 2020; Liu et al., 2021) model, we use pointwise convolution to encode relative positional information into each feature channel, avoiding complex trigonometric operations. Notably, experiment shows that simply adding PE to the LA module does not lead to a significant performance improvement.

**Feature preprocess.** We calculate the feature differences between neighboring points and center point, thereby highlighting important features in local region. With the coupling step, we process the input features as follows:

$$f' = f - f_{center} + Relu(BN(Conv_{1 \times 1}(p))), \tag{6}$$

where $Conv_{1 \times 1}$ denotes a mapping from coordinate dimensions to feature dimensions, and $f_{center}$ corresponds to the features of the central point in groups. Therefore, we can use Eq. 6 to couple positional features and capture salient changes in local features.

**Grouping strategy.** Excessive summation of points may result in large feature values that affect parameter convergence. On the other hand, the traditional grouping method is overly redundant in sampling, while DSConv only requires less points that can reflect the spatial structure for extracting geometric features, so we reduce the number of sampled points. Although this may lead to uneven sampling and produce different results between consecutive samplings, it significantly improves speed and resolves convergence issues. To enhance sampled point quality, we modify the sampling method: firstly, N points are sampled by ball-query (Qi et al., 2017b) or KNN, then Farthest Point Sampling (FPS) is used to resample N/4 points, ensuring consistent and representative sampling.

## 3.5 ARCHITECTURE

We adjust the original architecture (Qian et al., 2022), with the improvements primarily reflected in the XXL scale model. We refer to PointNeXt for model scaling and further increase the number of DSConvs in each stage. In DSConv-XXL, to fully leverage the advantages of DSConv, we remove the initial MLP stem and the first downsampling, directly operating on the input point cloud, effectively increasing the number of processed points by $\times 4$. To handle the large point cloud data, we adopt grid-sampling for faster downsampling, inspired by Thomas et al. (2019); Wu et al. (2022). Additionally, due to the increased complexity of feature extraction in DSConv, residual connections preprocess the original input with an MLP to better adapt to the feature changes. Furthermore, we retain the inverted bottleneck structure, using it after every two DSConv layers. We represent the channel of the initial embedding MLP as C and the number of DSConv modules as B, and designed four sizes of models as follows:

- DSConv-B: C = 32, B = 1, 1, 2, 1
- DSConv-XL: C = 64, B = 3, 3, 6, 3
- DSConv-L: C = 32, B = 2, 2, 4, 2
- DSConv-XXL: C = 64, B = 4, 4, 8, 4

In segmentation tasks, we adopt the encoder-decoder structure of PointNet++ (Qi et al., 2017b). In classification tasks, only one encoder and one classification layer are used. For extremely large point cloud data, we use 5 stages for deeper downsampling. The overall architecture is depicted in Fig. 5.

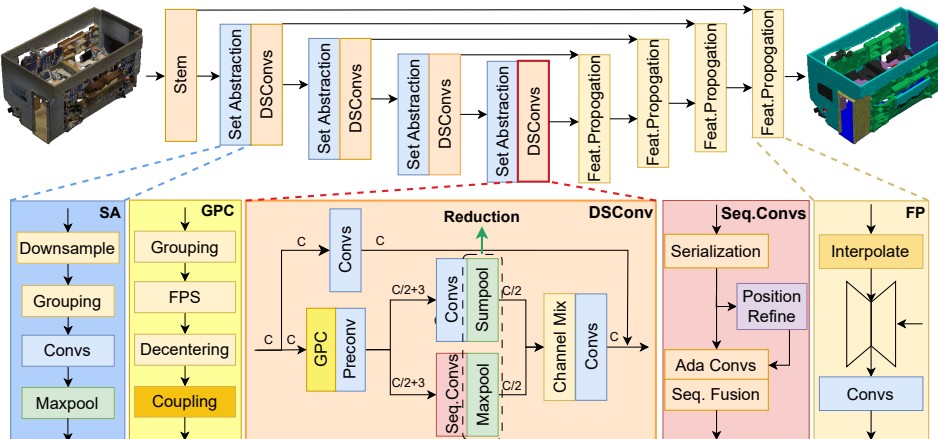

Figure 5: Overall Architecture. We retain the Set Abstraction (SA) module and Feature Propagation (FP) structure from PointNet++, and introduce the DSConv module. AdaConv refer to the adaptive convolution mentioned in Sec. 3.2&3.4, while Convs are 1×1 convolutions by default.

## 4 EXPERIMENTS

To validate the effectiveness of our approach, we conduct experimental evaluations on the semantic segmentation datasets ScanNet V2 (Dai et al., 2017a) and S3DIS (Armeni et al., 2016), as well as the point cloud classification dataset ScanObjectNN (Uy et al., 2019). To ensure fair contrast, except for the hyperparameters of the method proposed in this paper, we maintain the same data augmentation and hyperparameter settings as Qian et al. (2022), and use the same evaluation criteria. For the new DSConv-XXL, we use larger point cloud input and independent hyperparameter settings.

**Experimental setups.** We train DSConv using CrossEntropy loss with label smoothing (Szegedy et al., 2016), AdamW (Loshchilov & Hutter, 2019) optimizer, and cosine decay. For semantic segmentation task, we set the initial learning rate lr=0.01, weight decay $10^{-4}$, and train for 100 epochs. In classification task, DSConv is trained with an initial lr=0.002, weight decay of 0.05, and the number of input points is set to 1024 for 400 epochs. For DSConv-XXL, we set lr to 0.006 and warmup epoch to 10, while employing dropout to enhance generalization. For segmentation tasks, the voxel size is set to 0.02m and 0.04m for ScanNet V2 and S3DIS. In ScanNet V2, we adopt the latest training strategy from Chen et al. (2023) and expand the input points to 80,000. For fair comparison, we utilize the testing scheme provided by PointNext (Qian et al., 2022) for our tests. Model parameter size and GFLOPs are provided, and we also compare the model inference speed using throughput (TP) following Qian et al. (2021b; 2022). Throughput for segmentation models is measured with 8×15000 points, while for classification models, it is measured with 32×2048 points. We evaluate model performance using an RTX 4070 12GB GPU.

### 4.1 3D SEMANTIC SEGMENTATION ON S3DIS AND SCANNET

**Segmentation on S3DIS.** S3DIS (Stanford Large-Scale 3D Indoor Spaces) (Armeni et al., 2016) comprises laser-scanned data from 271 rooms across six different large-scale indoor areas, encompassing 13 categories. Results for both Area 5 and 6-fold cross-validation are presented in Tab. 1 and Tab. 2. With more modules and updated training strategies, DSConv-XXL achieves 75.5% mIoU on S3DIS Area 5 surpassing KPConvX by 2.0% mIoU. By removing the stem MLP from DSConv-XXL, the number of points processed by DSConv in each stage increases, while the number of feature channels is halved. This reduces parameters but increases GFLOPs. On S3DIS 6-fold cross-validation, DSConv-XL also achieves state-of-the-art performance, with a 4.8% improvement in mIoU compared to PointNeXt and outperforming the state-of-the-art PointVector (Deng et al., 2023) by 1.3% mIoU, with faster inference speed. The performance of DSConv-XL matches Point Transformer V3 on S3DIS Area 5, surpassing it by 2.0% mIoU on S3DIS 6-fold cross-validation and achieving a higher throughput. Notably, Point Transformer V3 (Wu et al., 2024) is trained using PPT (Wu et al., 2023). Thus, we compare with its original training results to ensure fairness.

Table 1: Semantic segmentation on S3DIS Area 5. Only the best test results of all algorithms are reported. The methods are listed in chronological order. The highest scores are marked in bold.

| Method | mIoU (%) | mAcc (%) | OA (%) | Params. (M) | FLOPs (G) | TP (ins./sec.) |
|---|---|---|---|---|---|---|
| SparseUNet (Choy et al., 2019) | 67.7 | 73.1 | 90.1 | 37.9 | 1.43 | 52 |
| RepSurf (Ran et al., 2022) | 68.9 | 76.0 | 90.2 | 0.98 | 1.04 | - |
| PointNeXt (Qian et al., 2022) | 70.5 | 76.8 | 90.6 | 41.6 | 84.8 | 40 |
| PTv2 (Wu et al., 2022) | 72.6 | 78.0 | 91.6 | 12.9 | 305 | 13 |
| PointVector (Deng et al., 2023) | 72.3 | 78.1 | 91.0 | 24.1 | 58.5 | 37 |
| PointMetaBase (Lin et al., 2023) | 72.3 | 78.0 | 91.3 | 19.7 | 11.0 | 83 |
| Swin3D (Yang et al., 2023) | 72.5 | - | - | 23.6 | 50.6 | 6 |
| PTv3 (Wu et al., 2024) | 73.4 | 79.0 | 91.7 | 46.2 | 42.7 | 29 |
| LPFP (Han et al., 2024) | 73.5 | 78.7 | 92.0 | 31.2 | 24.2 | - |
| OA-CNNs (Peng et al., 2024) | 71.1 | - | 90.7 | 51.5 | 23.5 | 22 |
| KPConvX (Thomas et al., 2024) | 73.5 | 78.7 | 91.7 | 13.5 | - | - |
| DSConv-B(our) | 70.9 | 76.3 | 90.9 | 3.4 | 9.7 | 129 |
| DSConv-L(our) | 72.4 | 78.2 | 91.3 | 5.5 | 14.5 | 101 |
| DSConv-XL(our) | 73.5 | 78.8 | 92.1 | 27.3 | 74.3 | 40 |
| DSConv-XXL(our) | **75.5** | **81.2** | **92.7** | 23.5 | 99.6 | 26 |

Table 2: Semantic segmentation on S3DIS 6-fold. Table 3: Semantic segmentation on ScanNet V2.

| Method | mIoU (%) | OA (%) | Method | Val mIoU (%) | Params.(M) |
|---|---|---|---|---|---|
| RepSurf-U (Ran et al., 2022) | 74.3 | 90.9 | StratTrans (Lai et al., 2022) | 74.3 | 18.8 |
| CBL (Tang et al., 2022) | 73.1 | 89.6 | DeLA (Chen et al., 2023) | 75.9 | 8.0 |
| PointNeXt (Qian et al., 2022) | 74.9 | 90.3 | OctFormer (Wang, 2023) | 75.7 | 39.0 |
| PTv2 (Wu et al., 2022) | 73.5 | - | Swin3D (Yang et al., 2023) | 76.4 | 23.6 |
| PointVector (Deng et al., 2023) | 78.4 | 91.9 | AVS-Net (Yang et al., 2024) | 76.0 | 20.9 |
| PointMetaBase (Lin et al., 2023) | 77.0 | 91.3 | OA-CNNs (Peng et al., 2024) | 76.1 | 51.5 |
| Swin3D (Yang et al., 2023) | 76.9 | - | PTv3 (Wu et al., 2024) | 77.5 | 46.2 |
| PTv3 (Wu et al., 2024) | 77.7 | - | KPConvX (Thomas et al., 2024) | 76.3 | 13.5 |
| DSConv-XL(our) | **79.7** | **92.2** | DSConv-XXL(our) | **77.8** | 25.9 |

**Segmentation on ScanNet.** ScanNet V2 (Dai et al., 2017a) comprises scans in indoor rooms, divided into 1201 training scenes, 312 validation scenes, and 100 testing scenes, with a total of 20 categories. We use mIoU as the validation metric for ScanNet V2. In ScanNet V2, we add a DSConv stage at the beginning of architecture like Chen et al. (2023), and experiments show that this stage at higher resolution is beneficial. As shown in Tab. 1, DSConv-XXL achieves the state-of-the-art performance among all methods **without** extra training data and outperforms PTv3 by 0.3% mIoU.

## 4.2 3D OBJECT CLASSIFICATION ON SCANOBJECTNN

ScanObjectNN (Uy et al., 2019) consists of approximately 15,000 actual scanned objects with 15 categories. Similar to PointNeXt, we evaluate the challenging variant PB_T50_RS of ScanObjectNN, using the mean±std deviation of OA and mAcc as our metrics. In the classification model, we use max pooling for reduction. As shown in Tab. 4, our DSConv-S surpasses PointNeXt-S by 0.6% OA, and a 1% increase in mAcc demonstrates that our algorithm achieves a more balanced performance across different categories. Meanwhile, our DSConv-XXL achieves **a best result of 91.0% OA**, reaching the state-of-the-art performance.

Table 4: Classification on ScanObjectNN **without** pre-training.

| Method | OA(%) | mAcc(%) |
|---|---|---|
| PointNet++ (Qi et al., 2017b) | 77.9 | 75.4 |
| RepSurf-U (Ran et al., 2022) | 86.0 | 83.1 |
| PointMLP (Ma et al., 2022) | 85.4±1.3 | 83.9±1.5 |
| PointNeXt (Qian et al., 2022) | 87.7±0.4 | 85.8±0.6 |
| PointVector (Deng et al., 2023) | 87.8±0.4 | 86.2±0.5 |
| PointMetaBase (Lin et al., 2023) | 87.9±0.2 | 86.2±0.7 |
| SPoTr (Park et al., 2023) | 88.6 | 86.8 |
| DeLA (Chen et al., 2023) | 90.1±0.2 | 89.0±0.4 |
| KPConvX (Thomas et al., 2024) | 88.3±0.4 | 86.7±0.5 |
| DSConv-S(our) | 88.3±0.4 | 86.8±0.6 |
| DSConv-XXL(our) | **90.6**±0.2 | **89.6**±0.4 |

## 4.3 ABLATION STUDY

We conduct ablation studies on DSConv, focusing on the impact of algorithm design on accuracy and throughput. By default, we test DSConv-L on S3DIS (Armeni et al., 2016) Area 5, employing Sequential Position Refinement method. All training parameters are configured identically.

**Effectiveness of serialization.** To validate the effectiveness of our serialization method, we compare it with other serialization methods. The 3rd-order Z-order represents serialization with three different encodings, while random indicates no serialization. As both Z-order (Morton, 1966) and Hilbert (Hilbert & Hilbert, 1935) are encoding distance-based serialization strategies, we only test

Table 5: Effectiveness of serialization.

| Method | OA(%) | mAcc(%) | mIoU(%) |
|---|---|---|---|
| Random (w/o ser.) | 90.2 | 76.2 | 69.9 |
| Z-order | 90.5 | 77.0 | 70.8 |
| 3rd-order Z-order | 90.5 | 77.2 | 71.1 |
| Coordinate-Axis | **91.3** | **78.2** | **72.4** |

Table 6: Point cloud position refinement methods.

| Method | OA(%) | mAcc(%) | mIoU(%) | TP |
|---|---|---|---|---|
| w/o refine | 90.6 | 77.0 | 71.3 | **107** |
| PPR | 91.0 | 77.4 | 71.8 | 105 |
| D-SPR | **91.4** | 78.0 | 72.3 | 59 |
| SPR | 91.3 | **78.2** | **72.4** | 101 |

Table 7: Convolution shape and sequence fusion methods. $3\times1\times3$ (xyz) means using three $1\times3$ convolutions along the three sequences of x, y, z.

| Method | OA(%) | mAcc(%) | mIoU(%) | TP |
|---|---|---|---|---|
| MLPs | 90.9 | 77.3 | 71.2 | 105 |
| $3\times1\times5$ (xyz) | 90.5 | 77.2 | 71.0 | 96 |
| $2\times1\times3$ (xy) | 91.0 | 77.6 | 71.7 | **110** |
| $3\times1\times3$ (xyz) | **91.3** | **78.2** | **72.4** | 101 |
| Maxpool | 90.7 | 77.0 | 71.1 | 93 |
| Concat | 91.2 | 77.9 | 71.8 | 92 |
| Add | **91.3** | **78.2** | **72.4** | **101** |

Table 8: Module design ablation. FPS ensures that sequences have engough spatial coverage, so it is included in Seq.AdaConv.

| PPC | Global path | Seq.Adaconv | SPR | mIoU(%) |
|---|---|---|---|---|
| | | | | 69.0 |
| ✓ | | | | 69.3 |
| ✓ | ✓ | | | 69.5 |
| ✓ | ✓ | ✓ | | 71.3 |
| | ✓ | ✓ | ✓ | 69.8 |
| ✓ | | ✓ | ✓ | 70.7 |
| ✓ | ✓ | ✓ | ✓ | **72.4** |

Z-order. As shown in Tab. 5, our method proposed in Sec. 3.1 outperforms other serialization methods, indicating that our approach can better express relative positional relationships and is effectively utilized by convolution. The inferior performance of 3rd-order Z-order method compared to ours shows that the improvement of our algorithm is not solely due to an increase in model parameters.

**Point cloud position refinement methods.** We explore the effectiveness of different point cloud position refinement methods proposed in Sec. 3.3 and their impact on model throughput. As shown in Tab. 6, our SPR method achieves performance close to D-SPR (a SPR based on DCN method, which samples neighborhoods for each sequence to generate intermediate feature maps of size 9 × the original feature size), demonstrating a certain level of flexibility and significantly improving throughput. Despite having less flexibility, the PPR method still enhances performance. This result indicates that incrementally refining point cloud positions as needed during feature extraction is effective, and the higher flexibility, the more pronounced the performance improvement.

**Convolution strategies.** We evaluate different receptive field shapes and multi-sequence fusion algorithms. As shown in Tab. 7, the receptive field of $3\times1\times5$ is limited by the number of sampling points, resulting in a lower mIoU. In contrast, the $3\times1\times3$ receptive field achieves higher performance with fewer parameters and computational cost. Using BEV (Yang et al., 2018) perspective (xy sequences) for receptive fields performs poorly due to the loss of information in the Z-axis direction. Additionally, using different convolution parameters for sequences in different directions is beneficial. Using addition for sequence fusion has fewer parameters and a higher mIoU than others.

**Module design.** We analyze each modules introduced in DSConv: the position-parameter coupling (PPC), Local-Global structure, sequence adaptive convolution and point position refinement. The results are illustrated in Tab. 8. PointNeXt-L (Qian et al., 2022) is our baseline. Simply adding PPC or global path in baseline do not significantly improve performance. However, removing them from DSConv leads to a significant performance decline, indicating that sequence convolution relies on global information and is more sensitive to position. Additionally, it shows that the coupling of position and parameters enhances the flexibility of convolution. By gradually adding each component we propose, we improve the baseline results to 72.4% mIoU. The increase in mIoU demonstrates the effectiveness of each component. More experiments are provided in Appendix B.

## 5 CONCLUSION

In this paper, we introduce DSConv, which achieves state-of-the-art performance on S3DIS, ScanNetV2 and ScanObjectNN. We utilize coordinate-based serialization to preserve the relative positional information of point clouds, followed by feature extraction using convolution. Additionally, we propose adaconv and multi-sequence fusion strategies to achieve finer-grained sequential convolution. Local-global structure ensures the global features within the groups. Position refinement and parameter coupling enhances the flexibility in processing point clouds. By adjusting input features and optimizing sampling methods, our model still maintains excellent scalability and speed.

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

# A CLASSIFICATION ARCHITECTURE

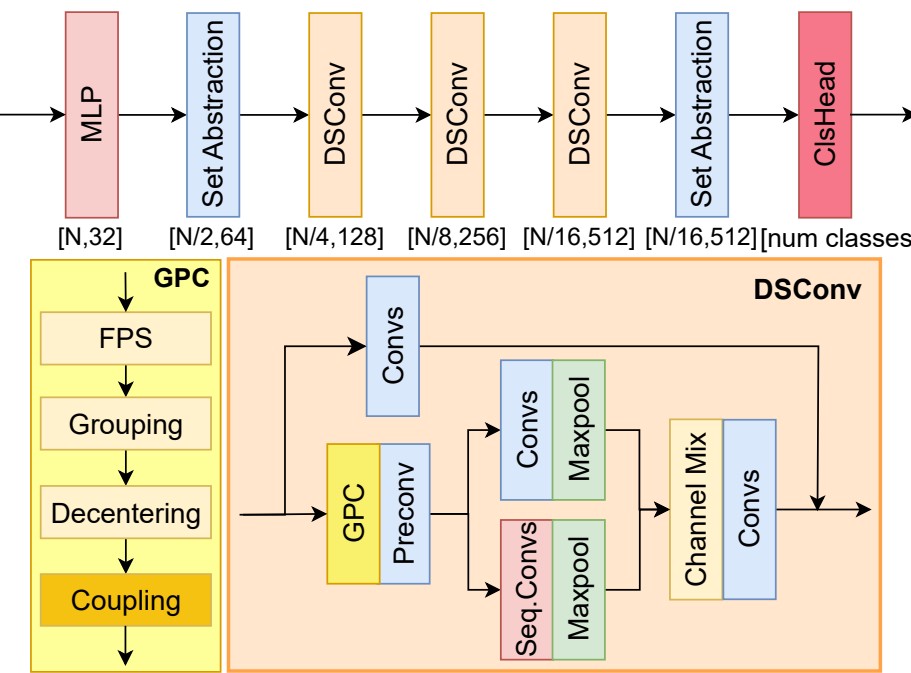

Figure 6: DSConv-S architecture for classification. We replace the SetAbstraction modules with DSConv modules and make adjustments to the DSConv modules, while the rest of the components remain unchanged.

As shown in Fig. 6, we replace the middle three SetAbstraction modules of PointNeXt with our DSConv modules. We keep the first SetAbstraction module to guarantee the accuracy of model, and maintain the last SetAbstraction module for its global pooling function. Meanwhile, to ensure the extraction of key points, we set the number of neighbor points to 32 and move the FPS block after grouping to the front for downsampling. Additionally, we swap the sum pooling in the MLP path with max pooling to improve performance in classification tasks. For DSConv-XXL, we design it according to Sec. 3.5 without changing SetAbstraction module, and reduce the arrangement of DSConvs to [2, 2, 2].

# B EXPERIMENTS

## B.1 3D SEMANTIC SEGMENTATION ON S3DIS

Table 9: S3DIS 6-fold cross-validation.

| Method | Metric | Area1 | Area2 | Area3 | Area4 | Area5 | Area6 | 6-Fold |
|---|---|---|---|---|---|---|---|---|
| PointVector-XL | OA | 92.69 | 91.14 | 93.81 | 90.24 | 91.01 | 94.27 | 91.86 |
| | mAcc | 89.58 | 79.07 | 92.22 | 83.81 | 78.09 | 92.84 | 86.12 |
| | mIoU | **82.38** | **70.56** | 84.64 | 70.17 | 72.29 | 86.40 | 78.41 |
| DSConv-L | OA | 91.96 | 89.05 | 93.74 | 89.89 | 91.27 | 94.36 | 91.41 |
| | mAcc | 88.02 | 75.38 | 92.86 | 82.86 | 78.21 | 92.37 | 85.46 |
| | mIoU | 80.88 | 65.94 | 84.99 | **72.46** | 72.43 | 86.76 | 77.99 |
| DSConv-XL | OA | 92.46 | 91.17 | 93.92 | 90.57 | 92.09 | 94.34 | 92.22 |
| | mAcc | 89.40 | 79.82 | 92.90 | 82.41 | 78.79 | 92.86 | 87.07 |
| | mIoU | 82.08 | 70.18 | **85.11** | 71.10 | **73.49** | **87.21** | **79.65** |

We introduce the detailed results on S3DIS with 6-fold cross-validation and compare them with the previous state-of-the-art method, PointVector (Deng et al., 2023). As shown in Tab. 9, our DSConv-XL surpasses PointVector in all three metrics: OA, mAcc, and mIoU, and it performs better in 4 areas, including an improvement of 1.2% mIoU on Area 5. Meanwhile, we observe the mIoU of DSConv-XL is 1.36% lower than that of DSConv-L on S3DIS Area 4. Further experiments indicate that DSConv-XXL performs with lower accuracy than DSConv-XL in some areas. Therefore, we select DSConv-XL as the best model on S3DIS 6-fold cross-validation.

### B.2 3D OBJECT CLASSIFICATION ON MODELNET40

ModelNet40 (Wu et al., 2015) is a commonly used dataset for 3D object recognition and classification, comprising 40 different categories, each containing unique 100 CAD models with varying orientations and poses. Due to the inclusion of multiple categories and a large number of 3D object models, the ModelNet40 dataset presents challenges for algorithm generalization and robustness. However, recent research has favored ScanObjectNN, which is based on real-world objects. Therefore, we mainly benchmark DSConv on ScanObjectNN. Here, our results on ModelNet40 are also provided. We employ the exact same training strategy as DeLA, including AdamW

Table 10: Classification on ModelNet40.

| Method | OA(%) | mAcc(%) |
|---|---|---|
| PointNet (Qi et al., 2017a) | 89.2 | 86.2 |
| PointNet++ (Qi et al., 2017b) | 91.9 | - |
| PointCNN (Li et al., 2018) | 92.2 | 88.1 |
| DGCNN (Wang et al., 2019b) | 92.9 | 90.2 |
| KPConv (Thomas et al., 2019) | 92.9 | - |
| ASSANet-L (Qian et al., 2021b) | 92.9 | - |
| MVTN (Hamdi et al., 2021) | 93.5 | **92.2** |
| CurveNet (Xiang et al., 2021) | 93.8 | - |
| PTv1 (Zhao et al., 2021) | 93.7 | 90.6 |
| PointMLP (Ma et al., 2022) | 94.1 | 91.3 |
| PTv2 (Wu et al., 2022) | **94.2** | 91.6 |
| PointNeXt (Qian et al., 2022) | 94.0 | 91.1 |
| DeLA (Chen et al., 2023) | 94.0 | 92.1 |
| DSConv-XXL(our) | 94.0 | **92.2** |

optimizer, a learning rate of 0.002, cosine learning rate decay, weight decay of 0.05, a batch size of 32 for 600 epochs, and 1024 input points, with random scaling and translation as data augmentation. As shown in Tab. 10, our algorithm achieves results similar to DeLA (Chen et al., 2023) and PointNeXt (Qian et al., 2022). Furthermore, consistent with the ScanObjectNN test, the weighted sum downsampling of convolution does not effectively extract crucial edge features, which is the primary reason for the lack of greater performance improvement in our algorithm.

### B.3 CONTRAST WITH 3D CONVOLUTION-BASED METHODS.

Table 11: Contrast with 3D convolution-based methods upon PointNeXt baseline on S3DIS Area 5.

| Method | OA (%) | mIoU (%) | Params. (M) | TP (ins./sec.) |
|---|---|---|---|---|
| PointConv (Wu et al., 2019) | 89.5 | 67.9 | 29.9 | 30 |
| KPConv (Thomas et al., 2019) | 89.9 | 69.5 | 17.3 | 54 |
| PAConv (Xu et al., 2021) | 90.7 | 70.5 | 27.3 | 31 |
| DSConv(our) | **91.3** | **72.4** | **5.5** | **101** |

To validate the effectiveness of our serialization method and adaptive convolution, we conduct a comparative analysis with 3D convolution-based methods. To compensate for potential shortcomings in the training strategies of previous 3D convolution-based methods and to ensure a fair comparison, all methods are trained and evaluated under the PointNeXt-L (Qian et al., 2022) baseline by default, employing identical training and testing parameters. Similar to DSConv, we solely replace the LA module of PointNeXt with other 3D convolution-based methods, while preserving the residual structure in PointNeXt. Furthermore, we adopt the same block stacking strategy. For speed test, $16 \times 15,000$ points are used to measure throughput. As shown in Tab. 11, our method demonstrates a significant advantage among all approaches applying 2D convolutions to 3D point clouds, achieving an accuracy improvement of 1.9% mIoU over PAConv (Xu et al., 2021). Moreover, our method significantly outperforms other convolution-based methods in terms of speed, primarily due

to the combination of serialization and AdaConv, which effectively avoids the complexities of spatial mapping in point clouds. Additionally, DSConv has only 5.5M parameters, because we directly compute the point cloud sequences using 2D convolutions, rather than employing a large number of MLPs to simulate convolution, as seen in previous methods.

### B.4 ABLATION STUDY

**MLPs in DSConv module.** In addition to convolution, our DSConv module also employs MLPs to accomplish feature fusion at the front-end and terminal positions. To validate the necessity of these two parts of MLPs, we conduct ablation studies on DSConv-L, testing various combinations of MLPs. Preconv and Postconv are located after the positional encoding and before the summation of residual connections, respectively. They are respectively used to fuse in-put features with positional encoding and integrate features from MLP path and sequential con-volution path. Both Preconv and Postconv are composed of point-wise convolutions. In addition, Attention refers to the use of a channel attention mechanism to process the input features. As shown in Tab. 12, using a simple MLPs structure for positional encoding fusion is more effective. Furthermore, since the DSConv module introduces multi-layer convolutions during feature processing, the original use of bottleneck convolutions would excessively increase the network depth, affecting the performance of the residual structure. Therefore, using a single MLP layer in Postconv achieves higher mIoU and OA.

Table 12: The results of different MLP combinations for feature fusion on S3DIS Area 5.

| Method | OA(%) | mAcc(%) | mIoU(%) |
|---|---|---|---|
| Preconv | 91.3 | 78.2 | 72.4 |
| Preconv+Attention | 90.6 | 75.8 | 70.2 |
| Attention | 90.3 | 76.9 | 70.5 |
| w/o Pre-MLPs | 90.8 | 77.4 | 71.6 |
| Postconv | 91.3 | 78.2 | 72.4 |
| Bottleneck Postconv | 90.8 | 77.0 | 71.0 |
| w/o Post-MLPs | 90.5 | 76.0 | 70.3 |

**Skip connection.** We experiment with different residual structures. Concat+Mlp refers to concatenating the output of the cross-layer connection with the result of the main path, and then fusing them through a MLP. As shown in Tab. 13, the method of fusing the main path and the skip connection through Concat+MLP outperforms the residual structure (He et al., 2016) of PointNeXt. But using only linear for the skip connection achieves better results. Therefore, we choose it as our baseline.

Table 13: Skip connection in DSConv module.

| Method | OA(%) | mAcc(%) | mIoU(%) |
|---|---|---|---|
| Identity | 91.0 | 76.8 | 70.8 |
| Concat+MLP | 91.1 | 76.2 | 71.1 |
| Linear | 91.3 | 78.2 | 72.4 |

## C VISUALIZATION

As shown in Fig. 7, our algorithm is able to perform better at the boundary planes, with more distinct segmentation lines for planar regions. This indicates that our algorithm is more sensitive to edge information, similar to 2D convolution algorithms. The utilization of local-global features gives us an advantage in both large complex scenes and simple scenarios. However, like most methods, our algorithm also exhibits limitations in segmenting small embedded components, such as small windows on walls.

## D EFFECTIVENESS OF DSCONV

We visualize output features of the last module in DSConv using Grad-CAM (Selvaraju et al., 2017) and compare it with KPConv to analyze the effectiveness of DSConv. As shown in Fig. 8, DSConv demonstrates a higher sensitivity to edge features, with a greater distinction between high and low attention regions. This is because DSConv performs convolution within serialized group points, allowing it to extract more fine-grained features. Additionally, the high flexibility of DSConv make it easier to learn complex shapes, such as airplane tail fins and computer brackets.

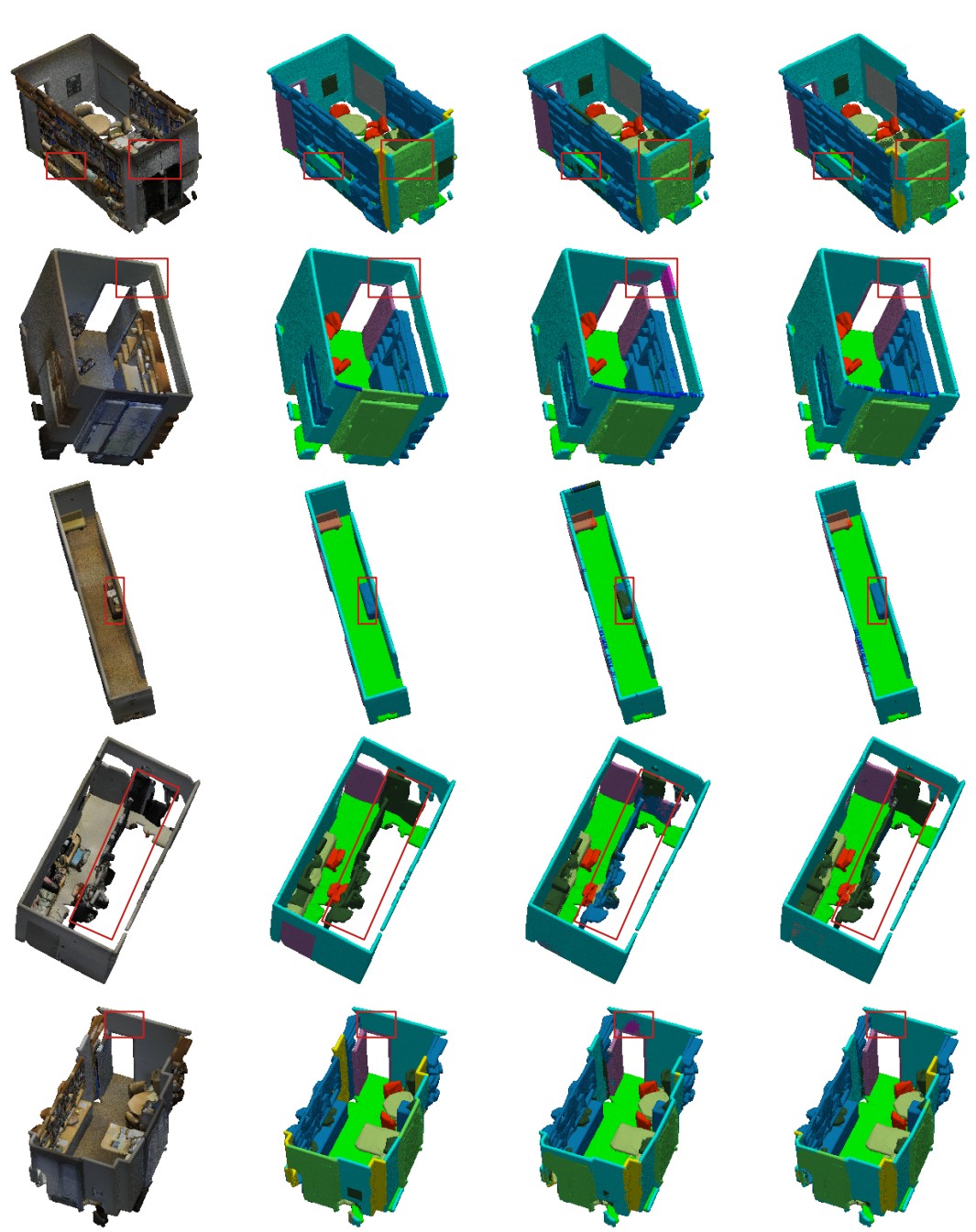

Figure 7: Qualitative comparisons of Ground Truth ($2^{nd}$ column), PointNeXt ($3^{rd}$ column), and DSConv ($4^{th}$ column) on S3DIS semantic segmentation. The specific differences are highlighted with red anchor boxes.

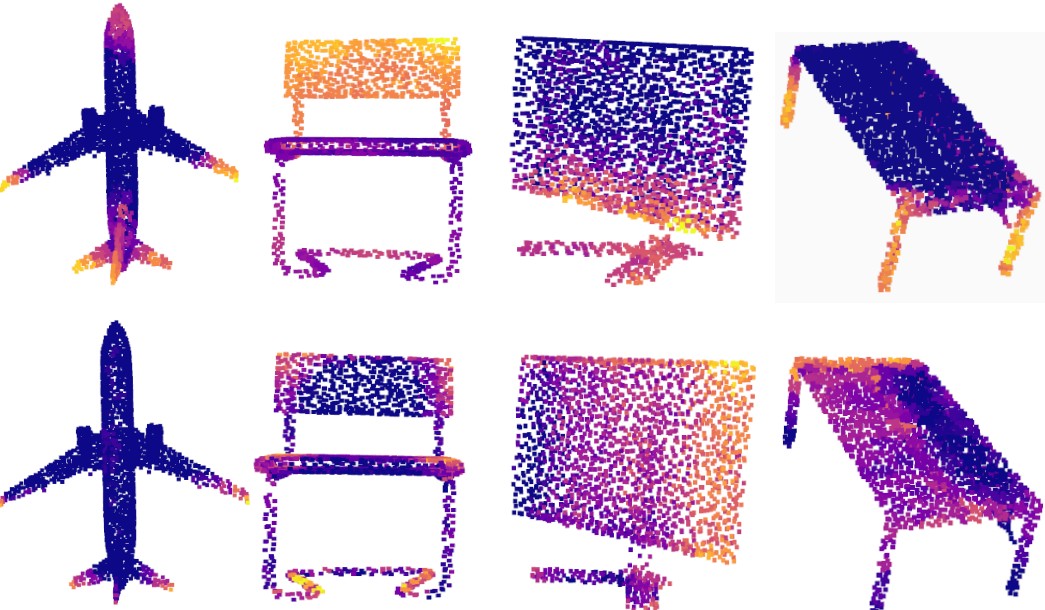

Figure 8: Attention visualization results of DSConv ($1^{st}$ row) and KPConv ($2^{nd}$ row) on Model-Net40. High activations are in yellow and low activations in blue.

