# OpenReview forum: "DSConv: Dynamic Convolution On Serialized Point Cloud"
_ICLR.cc/2025/Conference — ICLR 2025 Conference Withdrawn Submission_

### Official Review · Reviewer_9WDZ · 2024-10-28

**Soundness:** 2
**Presentation:** 3
**Contribution:** 2
**Rating:** 5
**Confidence:** 4

**Summary:**

The paper introduces a method, DSConv (Dynamic Sequence Convolution), designed to address the challenges of unstructured point clouds in 3D data analysis. Point clouds, commonly used in 3D object classification and semantic segmentation, lack structural consistency, making it difficult to apply traditional convolutions. DSConv employs serialization to impose a structured sequence onto point clouds, allowing efficient convolution operations. The proposed method incorporates Adaptive Convolution (AdaConv), dynamic point position refinement, and a combination of local-global feature extraction to achieve high accuracy with reduced computational overhead. DSConv is evaluated on datasets such as ScanObjectNN, ScanNet V2, and S3DIS, where it shows superior performance over state-of-the-art methods in both accuracy and speed.

**Strengths:**

1.	Innovative Approach: DSConv introduces a novel approach to handle unstructured point clouds by serializing them, enabling efficient and scalable convolution operations.
2.	Efficiency: DSConv achieves improvements in computational efficiency, as demonstrated by higher throughput metrics. The use of a coordinate-based serialization method and simplified dynamic position refinement contribute to this efficiency.
3.	Comprehensive Experiments: Extensive experimentation and ablation studies validate the effectiveness of the proposed methods and design choices. The model achieves state-of-the-art results on multiple challenging datasets.
4.	Writing: This paper is organized and written well.

**Weaknesses:**

1.	Limitations in Serialization: Although DSConv’s coordinate-axis-based serialization maintains relative positional relationships, it may not fully capture neighboring relationships in very sparse or highly dense point clouds. Additionally, for objects with complex geometries, the serialization can be inconsistent, potentially leading to unstable local feature extraction.
2.	Computational Cost of Dynamic Position Adjustment: Despite optimizations, the dynamic point position refinement in DSConv still adds computational overhead, especially in large-scale point cloud datasets. This can limit the efficiency of DSConv in memory-constrained environments, where dynamic adjustments can be resource-intensive.
3.	Scalability across Object Sizes: DSConv relies on fixed convolution and serialization parameters, which may not be ideal for objects with significant variations in scale. Small-scale details may be lost with a fixed convolution kernel size, potentially impacting the model’s ability to capture finer features in smaller objects.
4.	Robustness to Noise and Deformation: DSConv may be less robust to noise and deformation in point cloud data than graph convolutional or deep attention-based models. The reliance on serialization and local feature extraction can make DSConv more sensitive to irrelevant features introduced by noise, potentially reducing its reliability in unstructured, real-world point clouds.
5.	Limited Comparison about efficiency on ScanobjectNN: The paper lacks a detailed comparison of parameters and latency with other methods on the ScanObjectNN dataset. It will be interesting to show the comparison.
6.	Limited Comparison with other method: in Table.1, lack the comparison with DeLA[1].
[1] Decoupled local aggregation for point cloud learning.

**Questions:**

Refer to the Weakness. I will improve my rates when the weaknesses are solved.

---

### Official Review · Reviewer_56yh · 2024-11-02

**Soundness:** 3
**Presentation:** 3
**Contribution:** 2
**Rating:** 5
**Confidence:** 5

**Summary:**

This paper proposes a dynamic convolution method called AdaConv, based on serialized point clouds. It utilizes relative positions among local point neighbors to perform feature grouping. Additionally, a dynamic point position refinement is introduced to dynamically modify the point positions according to object pose variation. Moreover, a local-global network structure is developed. The experiments are conducted on S3DIS and ScanNet for semantic segmentation, and ScanObjectNN for object classification.

**Strengths:**

1. Good performance.
2. The code is provided in the supplementary material.
3. The paper is well-organised.

**Weaknesses:**

1. The proposed method is incremental, it is a fusion of **lots of highly engineered modules**, *without* a clear centralized key idea or motivation. Concretely, it involves two main modules, AdaConv and dynamic refinement. AdaConv is heavily based on existing point cloud serialization techniques, where it only modifies the center point parameters and proposes an extra multi-sequence fusion. As for the local-global network structure, it further proposes **4** different techniques in Sec 3.4.
2. The paper writing is very confusing, without a clear/explicit description. There is no logical link between all three main contributions, making all the main contribution seems just like an independent engineering technique. Also, some statements are not reasonable. For example, Line 37 argues that 2D is better than 3D for capturing fine details, which is not reasonable.
3. How can the proposed position refinement help to handle rotation and deformation as stated in paragraph 1. There is no straightforward experiment to verify this. For example, the pose perturbation test.
4. The result on ScanNet (which is more challenging and important than S3DIS) is not impressive, only 0.3% higher than PTv2, although more efficient.
5. Some related works [1, 2] are not discussed.

*Refs*:
[1] Relation-Shape Convolutional Neural Network for Point Cloud Analysis. CVPR 2019.
[2] PAConv: Position Adaptive Convolution with Dynamic Kernel Assembling on Point Clouds. CVPR 2021.

*Justification*:
As there have been many SOTA fully-supervised point cloud learning methods in recent years, this paper does not show its specialty.
It is very incremental without clear motivation, and the performance improvements mainly come from the highly engineered techniques, instead of a novel/insightful idea, as stated in the aforementioned weaknesses. As a result, the reviewer thinks that this paper may be a **good project** but is **not qualified for a good scientific paper**.

**Questions:**

The author should really think about one question: "what is the key idea of this paper".

---

### Official Review · Reviewer_ZHYK · 2024-11-03

**Soundness:** 3
**Presentation:** 3
**Contribution:** 2
**Rating:** 6
**Confidence:** 4

**Summary:**

This paper introduces coordinate-based serialization to preserve relative positional information among point clouds. A corresponding Dynamic Sequence Convolution module is proposed for feature aggregation.

**Strengths:**

1. This paper introduces an axis-group serialization different from previous techniques used in existing methods, such as Point Transformer V3.
2. The proposed pipeline achieves good performance on various datasets, outperforming rececnt SOTAs.

**Weaknesses:**

The proposed method outperforms previous baselines by a large margin. Though not significant, this paper does introduce a few contributions, including serialization (used in PointGPT [1], Point Transformer V3 [2], etc), local-global structure (PointNet [3])


[1] Chen, Guangyan, et al. "Pointgpt: Auto-regressively generative pre-training from point clouds." Advances in Neural Information Processing Systems 36 (2024).
[2] Wu, Xiaoyang, et al. "Point Transformer V3: Simpler Faster Stronger." Proceedings of the IEEE/CVF Conference on Computer Vision and Pattern Recognition. 2024.
[3] Qi, Charles R., et al. "Pointnet: Deep learning on point sets for 3d classification and segmentation." Proceedings of the IEEE conference on computer vision and pattern recognition. 2017.

**Questions:**

I appreciate the comprehensive experiments across various datasets. The authors claim that the proposed method can maintain scalability and inference speed. Compared to Point Transformer V3, DSConv-XL, DSConvXXL seem to consume more memory to achieve better performance.

---

### Official Review · Reviewer_u6fQ · 2024-11-05

**Soundness:** 3
**Presentation:** 1
**Contribution:** 3
**Rating:** 3
**Confidence:** 4

**Summary:**

This paper proposes a serialization approach and AdaConv to improve point cloud deep networks.

**Strengths:**

The paper showed strong experiment results in multiple datasets.

The ablations on the serialization and SPR are convincing.

**Weaknesses:**

Overall, my main concern of this paper is that it is too badly written to the extent that most of the approaches and methods in the paper are not well-defined. One CANNOT define a method by a figure, this is not a scientific approach. Unfortunately most of the methods in the paper are defined on figures and even when there are equations, the notations were too bad to be a proper paper. Hence, I believe this paper requires a major revision before it can be accepted.

The proposed contribution, Axis and Axis-Group serialization, is very unclearly described. I can maybe figure out what is axis serialization (serializing along a single axis), but how does Axis-Group serialization work? The red circle drawn is centered at one point and how do different axes and different groups come into play to provide the serialization of the entire point cloud? This approach requires a better description which is lacking in the current manuscript.

I am confused on what 2D convolution is being applied in the paper. The equations (1) and (2) seem to be describing 1D convolutions.

The \times operator is not defined and I don't believe it is a normal multiplication (looks more similar to an inner product). It would be better to use standard notations if it is an inner product.

After serialization, the point features should be in the form of B x N x D, with B being the batch size, N being the number of points and D being the dimensionality. It's a 3-dimensional tensor. I don't understand why it's a form similar to a four-dimensional tensor of images because there is one less dimension in there.

The notations in Sec. 3.3 are again problematic. It is unclear whether features are involved in the part called AdaConv(\delta p_i) and the English is problematic as well such as sentences like the sentence that starts with "The original input is computed using AdaConv^{p+f}..."

For the PPR approach, the text said only the center point is modified but the figure looks like the center stayed the same and all the other points were modified.

Inference speed is measured with too few points, it's not reflective of realistic inference speeds on e.g. ScanNet where there would be 50k - 150k points.

**Questions:**

Please work on the weaknesses and try to better write the paper.

---

### Official Review · Reviewer_YPrJ · 2024-11-07

**Soundness:** 1
**Presentation:** 1
**Contribution:** 1
**Rating:** 3
**Confidence:** 4

**Summary:**

The paper presents a model architecture to process point clouds based on a convolution operation working directly on points. The method serialize the positions of the points in the convolution receptive field and uses 2D convolution to aggregate the features of neighboring points. Additionally, the model also uses such convolutions to compute a point displacement and modify the point coordinates in order to obtain more stable features.

**Strengths:**

- The paper explores a new neural architecture based on points instead of voxels.
- The experiments presented in the paper show similar performance to SOTA method based on sparse voxels and transformers.
- Most of the design choices are ablated on the experiments section.

**Weaknesses:**

I believe the main weaknesses of the paper are the novelty, motivation, missing experiments, clarity, and marginal improvements. In the following paragraphs, I elaborate each of them in detail:

- Novelty: The idea of rearranging the points in the neighborhood and using a fixed set of weights is not new. One of the first papers in the field PointCNN, already perform this in order to design their convolution. The main difference is that in their paper they learned this transformation while in this one the transformation is given by a simple serialization on the x,y, and z coordinates. Additionally, the other contribution, computing a displacement of the point coordinates with a convolution was initially proposed by KPConv.

- Motivation: The motivation of the serialization of a local receptive field is unclear. The serialization methods z-order or Hilbert are specifically designed to transform 3D space into a 1D sequence where neighboring points in 3D space are near in the 1D sequence. So the motivation of using the ball query to avoid large jumps in 3D space defeats the purpose of using a serialization method since this will only happen in a few specific cases (especially if a Hilbert curve is used). Moreover, the paper indicates the serialization methods do not preserve the relative position of points, but the proposed serialization method also does not. The serialization method is used to select the close points, while the relative position is processed by the features (as it is done in the proposed convolution). In the case of Octformer of PointTransformerv3, the relative position information is processed by a sparse convolution + Attention block.

- Missing experiments: A key experiment is missing, evaluation of the model with Hilbert serialization. This method preserves locality much better than z-order and should be evaluated accordingly.

- Clarity: The paper is very difficult to follow and the proposed methods are unclear:
	- The serialization method is never formally defined.
	- What is w in Formula 1? The convolution is never defined before this point and the reader does not know what is w.
	- The motivation for the dynamic position refinement is unclear. Why is this necessary? What does this sentence mean? "The objects in three-dimensional space exhibit richer geometric transformations. To reduce the impact of these intricate pose variations on feature extraction, ..."
	- It is not clear what coordinates are modified in the dynamic position refinement or how those are supervised.
	- Figure 4 is unclear. The reader does not understand why or what is happening to the points.

- Weak experimental results: The improvements compared to other methods are marginal in almost all experiments and ablations.

**Questions:**

See weaknesses.

---

### Note · Authors · 2024-11-14

I have read and agree with the venue's withdrawal policy on behalf of myself and my co-authors.